# Effects of Hydrogen Sulfide on Carbohydrate Metabolism in Obese Type 2 Diabetic Rats

**DOI:** 10.3390/molecules24010190

**Published:** 2019-01-06

**Authors:** Sevda Gheibi, Sajad Jeddi, Khosrow Kashfi, Asghar Ghasemi

**Affiliations:** 1Endocrine Physiology Research Center, Research institute for Endocrine Sciences, Shahid Beheshti University of Medical Sciences, Tehran 19395-4763, Iran; sevda.1365@yahoo.com (S.G.); sajad.jeddy62@gmail.com (S.J.); 2Department of Molecular, Cellular and Biomedical Sciences, Sophie Davis School of Biomedical Education, City University of New York School of Medicine, New York, NY 10031, USA

**Keywords:** diabetes, H_2_S, glucose tolerance, blood pressure

## Abstract

Hydrogen sulfide (H_2_S) is involved in the pathophysiology of type 2 diabetes. Inhibition and stimulation of H_2_S synthesis has been suggested to be a potential therapeutic approach for type 2 diabetes. The aim of this study was therefore to determine the effects of long-term sodium hydrosulfide (NaSH) administration as a H_2_S releasing agent on carbohydrate metabolism in type 2 diabetic rats. Type 2 diabetes was established using high fat-low dose streptozotocin. Rats were treated for 9 weeks with intraperitoneal injections of NaSH (0.28, 0.56, 1.6, 2.8, and 5.6 mg/kg). Serum glucose was measured weekly for one month and then at the end of the study. Serum insulin was measured before and after the treatment. At the end of the study, glucose tolerance, pyruvate tolerance and insulin secretion were determined and blood pressure was measured. In diabetic rats NaSH at 1.6–5.6 mg/kg increased serum glucose (11%, 28%, and 51%, respectively) and decreased serum insulin, glucose tolerance, pyruvate tolerance and in vivo insulin secretion. In controls, NaSH only at 5.6 mg/kg increased serum glucose and decreased glucose tolerance, pyruvate tolerance and insulin secretion. Chronic administration of NaSH in particular at high doses impaired carbohydrate metabolism in type 2 diabetic rats.

## 1. Introduction

Hydrogen sulfide (H_2_S), a novel gasotransmitter, is involved in many (patho)physiological processes [1]. H_2_S exerts its well-known effect, i.e., decreasing blood pressure, mostly by opening ATP-sensitive potassium channels (K_ATP_) in vascular smooth muscle cells [2]. K_ATP_ channels also play a key role in insulin secretion in pancreatic β-cells [3], where opening of the channels by H_2_S decreases insulin secretion [4]. Both endogenous and exogenous H_2_S inhibit insulin secretion from β-cells by activation of K_ATP_ channels and inhibition of L-type voltage-dependent calcium channels [5]. In addition, through inhibition of glucose transporter-4 (GLUT-4), H_2_S inhibits insulin-stimulated glucose uptake in white adipocytes in rat, indicating that H_2_S decreases insulin sensitivity of adipocytes [6]. In Zucker diabetic fatty rats, the rate of endogenous H_2_S production is higher than non-diabetic rats and inhibition of H_2_S production restores hyperglycemia to near normal levels [7]. Inhibition of H_2_S synthesis has been therefore suggested to be a potential therapeutic approach for insulin resistance states and also for protecting β-cells during the induction phase of diabetes [8].

On the other hand, type 2 diabetes is a state of H_2_S deficiency [9,10,11,12] and H_2_S protects β-cells against glucotoxicity-induced apoptosis [13]. Chondroitin sulfate prebiotic diet, which increases H_2_S levels in colon, stimulates glucagon like peptide-1 (GLP-1) secretion, which has anti-diabetic actions [14]. H_2_S up-regulates insulin signaling pathways essential for glucose utilization [15] and mediates the effects of vitamin D on translocation of GLUT-4 in adipocytes [16]. These data are in favor of boosting H_2_S as a treatment for type 2 diabetes.

To our knowledge, there is only one in vivo study with the aim of assessing H_2_S effects on carbohydrate metabolism in type 2 diabetic rats [17]; results indicate decreased plasma glucose at a low dose of sodium hydrosulfide (NaSH) and increased plasma glucose at a high dose. Other studies, all involving db/db mice and assessing the effects of H_2_S administration on cardiac function [18,19,20] or learning and memory [21], have reported beneficial effects on glucose metabolism [20], no effect on blood glucose levels [18,19,21], or glucose tolerance [19,21].

Whether H_2_S has beneficial or deleterious effects on carbohydrate metabolism in type 2 diabetes has not been answered yet and before suggesting strategies to augment/inhibit H_2_S levels in clinic, further in vivo animal studies are needed considering long-term and dose-dependent effects of exogenous H_2_S. The aim of this study was therefore to determine effects of long-term administration of different doses of NaSH, a H_2_S donor, on carbohydrate metabolism in type 2 diabetic rats.

## 2. Results

### 2.1. Effect of NaSH on Serum Total Sulfide Levels

Diabetic rats had lower total serum sulfide levels than controls (16.5 ± 2.8 vs. 28.3 ± 2.4 µmol/L, *p* = 0.001). NaSH administration increased total serum sulfide levels in control rats at doses 2.8 and 5.6 mg/kg and in diabetic rats at doses 0.56 mg/kg and higher (Figure 1 and Appendix A).

### 2.2. Effect of NaSH on Body Weight, Water Consumption, and Food Intake

Body weights were comparable in all groups before starting the high fat diet (HFD). HFD consumption for 2 weeks caused a significant (*p* < 0.001) increase in body weights, but streptozotocin (STZ) injection to the HFD-fed rats resulted in a significant (*p* < 0.001) reduction in the body weights. Before starting the NaSH administration, the body weights were similar between all groups. During 9 weeks of NaSH administration, the body weights decreased gradually in the NaSH-treated diabetic rats and these decreases were significant at the doses of 2.8 and 5.6 mg/kg. The differences in body weight gain were significant from weeks 7 and 8 in NaSH-treated diabetic rats at 5.6 and 2.8 mg/kg, respectively, and were observed throughout the remainder of the study (Figure 2 and Appendix A).

As expected, in the diabetic rats, water consumption (Appendix A) and calorie intake (Appendix A) were significantly higher, while food intake (Appendix A) was significantly lower than the control group. NaSH-treated diabetic rats at 2.8 and 5.6 mg/kg displayed higher water consumption and lower food and calorie intake. NaSH administration in control rats had no effects on body weight, food and calorie intakes, or water consumption.

### 2.3. Effect of NaSH on Serum Glucose Concentration

Compared to the control group, diabetic rats had higher serum glucose levels before NaSH treatment (178.6 vs. 85.5 mg/dL, *p* < 0.001). After 9 weeks of NaSH administration at 1.6, 2.8, and 5.6 mg/kg, serum glucose concentrations increased significantly (253.0, 255.9, and 432.8 vs. 187.6 mg/dL, respectively, *p* < 0.001) and these became evident at weeks 4 (233.9 vs. 180.0 mg/dL, *p* = 0.001), 2 (247.3 vs. 184.9 mg/dL, *p* < 0.001), and 2 (292.4 vs. 184.9 mg/dL, *p* < 0.001), respectively. NaSH at 0.28 and 0.56 mg/kg, had no effect on serum glucose levels in diabetic rats. In the control rats, only the highest dose of NaSH (5.6 mg/kg) (128.3 vs. 82.5 mg/dL, *p* = 0.001) increased serum glucose concentrations at week 9 (Figure 3 and Appendix A).

### 2.4. Effect of NaSH on Serum Insulin Concentrations

Before starting NaSH treatment, serum insulin concentrations were comparable between the control and the diabetic rats (Appendix A). Continuing HFD increased serum insulin concentrations in non-treated diabetic rats (188.40 ± 14.09 vs. 79.31 ± 11.31 pmol/L, *p* < 0.001). In the diabetic rats, NaSH administration at 1.6, 2.8, and 5.6 mg/kg significantly decreased serum insulin levels (120.9 ± 14.3, 56.6 ± 8.7, and 34.5 ± 4.3 vs. 188.4 ± 14.1, pmol/L *p* < 0.001, respectively), whereas this was not changed at 0.28 and 0.56 mg/kg. In the control rats, only the highest dose of NaSH (5.6 mg/kg) significantly decreased serum insulin levels (50.4 ± 5.8 vs. 79.3 ± 11.3 pmol/L, *p* < 0.001) (Figure 4).

### 2.5. Effect of NaSH on Glucose Tolerance

As shown in Figure 5 and Appendix A, compared to the control rats, non-treated diabetic rats had impaired glucose tolerance (*p* < 0.001). Chronic NaSH administration at 1.6, 2.8, and 5.6 mg/kg, impaired glucose tolerance in diabetic rats (*p* < 0.001), while improved it at 0.56 mg/kg (*p* = 0.009) and had no effect at 0.28 mg/kg. In the control rats, only the highest dose of NaSH impaired glucose tolerance (*p* < 0.001).

### 2.6. Effect of NaSH on Gluconeogenesis

To investigate the ability of NaSH treatment to modulate gluconeogenesis, pyruvate tolerance test (PTT) was performed. In non-treated diabetic rats, pyruvate administration, a gluconeogenic substrate precursor, caused higher glucose production compared to the control rats (AUC: 26844 ± 1286.0 vs. 10523 ± 435.1, *p* < 0.001). Chronic NaSH administration at 1.6 (AUC: 30876 ± 1848 vs. 26844 ± 1286, *p* = 0.048), 2.8 (AUC: 40241 ± 2955 vs. 26844 ± 1286, *p* < 0.001), and 5.6 (AUC: 62190 ± 2303 vs. 26844 ± 1286, *p* < 0.001) mg/kg, increased serum glucose concentration during PTT in diabetic rats, while it had no effect at 0.28 and 0.56 mg/kg. In controls, NaSH only at 5.6 mg/kg significantly (AUC: 14598 ± 730.2 vs. 10523 ± 435.1, *p* = 0.046) impaired pyruvate tolerance (Figure 6 and Appendix A).

### 2.7. Effect of NaSH on In Vivo Insulin Secretion

We next measured serum insulin levels during glucose tolerance test (GTT) to determine the effects of NaSH on in vivo insulin secretion. In non-treated diabetic rats, increases in insulin secretion during GTT, were lower than the control rats (AUC: 29582 ± 2235 vs. 46553 ± 1857, *p* < 0.001); NaSH administration for 9 weeks at 1.6 (AUC: 21459 ± 2938 vs. 29582 ± 2235, *p* = 0.026), 2.8 (AUC: 17668 ± 3085 vs. 29582 ± 2235, *p* = 0.002), and 5.6 (AUC: 8112 ± 1000 vs. 29582 ± 2235, *p* < 0.001) mg/kg, decreased insulin response to glucose injection in the diabetic rats. In controls, NaSH only at 5.6 mg/kg significantly (AUC: 39150 ± 3323 vs. 46553 ± 1857, *p* = 0.028) decreased insulin response to glucose injection (Figure 7 and Appendix A).

### 2.8. Effect of NaSH on Blood Pressure

As shown in Figure 8, compared to the controls, non-treated diabetic rats had higher systolic blood pressure (SBP) (129.0 ± 1.9 vs. 112.6 ± 2.1, *p* = 0.007); NaSH administration for 9 weeks at all doses decreased SBP in diabetic rats. In controls, NaSH at 1.6, 2.8, and 5.6 mg/kg significantly decreased SBP.

## 3. Discussion

This study showed that chronic administration of high doses of NaSH aggravated carbohydrate metabolism in obese type 2 diabetic rats. NaSH administration at high doses increased serum glucose and impaired glucose tolerance and pyruvate tolerance, as well as decreased insulin secretion and serum insulin levels.

In this study, NaSH administration increased total sulfide levels in the serum of both control and diabetic rats, however, this increase was about 40% lower in the diabetic rats as compared to the corresponding controls. In this regard, literature reports are quite controversial as increased [22,23], decreased [10,18,24], and even no change [25] in blood H_2_S levels have been reported in diabetic patients [10,22,24] and animals [18,23,25]. Our results may be explained in light of what has been reported in the literature where it has been shown that hyperglycemia results in an increase in H_2_S consumption due to increases in reactive oxygen species [26]; moreover, the activities of H_2_S-generating enzymes have been reported to be lower in diabetes [18] which results in lower H_2_S levels which is in line with our observations reported here.

In this study, chronic NaSH administration in diabetic rats only at higher doses (2.8 and 5.6 mg/kg) inhibited weight gain, decreased food intake, and increased water consumption; NaSH administration, had however no effect on these parameters in the control rats. Few studies have reported effects of H_2_S on body weight gain in type 2 diabetes; in line with our results in control rats and low dose of NaSH in diabetic rats, no change in body weight have been reported following H_2_S donors administration in type 2 diabetic animals [18,20,21]. In addition, there was no significant differences in food intake in cystathionine γ-lyase-knockout (CSE-KO) mice [27] as well as in HFD-induced insulin resistance mice treated with the slow H_2_S-donating compound, GYY4137 [28].

In the present study, higher doses (1.6, 2.8, and 5.6 mg/kg) of NaSH increased fasting serum glucose and decreased serum insulin levels in the diabetic rats. Few studies have assessed the effects of H_2_S on glucose metabolism in type 2 diabetes; in line with our study, administration of high dose of NaSH (6.72 mg/kg for 6 weeks) to Goto-Kakizaki (GK) rats increased plasma glucose levels by 86.5% [17] (cf. with 115% increase in our study following 9 weeks of NaSH at 5.6 mg/kg).

In our study, diabetic rats exhibited insulin resistance as confirmed by 137% increase of fasting serum insulin in the presence of moderate stable hyperglycemia. Administration of NaSH dose-dependently decreased serum insulin levels in the presence of aggravated hyperglycemia, indicating decreased insulin secretion by pancreatic β-cells as previously reported in in vitro studies [4,29]. This result is also in agreement with decreased in vivo insulin secretion during GTT observed in our study. In our study, the highest dose of NaSH increased serum glucose and decreased serum insulin levels and in vivo insulin secretion in control rats. This is consistent with pathophysiology of type 2 diabetes with partial dysfunction of β-cells. H_2_S-induced hyperglycemia is associated with decreased glucose-induced Ca^2+^ influx [29], activating pancreatic β-cell K_ATP_ channels [4], and decreased ATP production in the pancreatic β-cells [29]. Other explanations for H_2_S-induced hyperglycemia include a decrease in peripheral glucose uptake and an increase in glycogenolysis [30].

In contrast to our results, Rong et al. reported a decrease in serum glucose concentrations following administration of NaSH at 1.68 mg/kg in GK type 2 diabetic rats [17]. Our model of obese type 2 diabetes (HFD-STZ) is characterized with a stable hyperglycemia (~15% fasting serum glucose fluctuation during 8 weeks) along with insulin resistance. However, non-obese GK rats in the study of Rong et al. did not show stable hyperglycemia and their fasting plasma glucose levels fluctuated by as much as 75% during the 10 weeks of their study. In addition, in the Rong et al. study, fasting plasma insulin at the end of their study was approximately one-third of the corresponding control rats, indicating the absence of insulin resistance as the main characteristic of type 2 diabetes. Indeed, it has been reported that β-cell defect is the primary defect in the GK rats [31]. These observations strongly suggest that the blood-glucose lowering effects of a low dose of H_2_S as has been reported by Rong et al. needs further verification in other models of type 2 diabetes.

Other studies that assessed the effects of H_2_S donors on blood glucose levels in type 2 diabetic animals are all in male db/db mice and have reported no effect on blood glucose concentrations using NaSH (4.5 mg/kg for 8 weeks) [21], Na_2_S (0.1 mg/kg for one week) [18], and NaSH (5.6 mg/kg for 12 weeks) [19]. These data showing that high doses of NaSH have no effect of on blood glucose levels is in contrast to our results reported here. One potential explanation for this apparent discrepancy is the use of different species. In our study we used rats whereas others used mice. The lethal dose (LD_50_) of NaSH following intraperitoneal (IP) injection has been reported to be lower in male rats (i.e., 15 mg/kg) [32] compared to male mice (i.e., 25.2 mg/kg) [33]. In addition, mouse models may not be suitable for studying H_2_S effects, as has been reported in assessing neurological squeal of H_2_S [34]; this however remains to be determined as it pertains to the metabolic effects of H_2_S.

In our study, NaSH at high doses impaired glucose tolerance and increased gluconeogenesis (as measured by PTT) in diabetic rats. Unlike our results, no change [21] or even an increase [17,20] in glucose tolerance have been reported in db/db mice [20,21] and GK rats [17] following NaSH administration. In our study, decreased insulin secretion in response to exogenous glucose resulted in poor glucose clearance during GTT. Our study is the first to show increased gluconeogenesis in type 2 diabetic rats; this finding is in line with both in vitro and in vivo studies in non-diabetic conditions. In CSE-KO mice, glucose production in hepatocytes is lower [30] and NaSH administration (2.18 and 3.53 mg/kg) increases gluconeogenesis [35]. H_2_S donors increased gluconeogenesis in vitro by increasing activities of gluconeogenic enzymes [30,35].

In the present study, all doses of NaSH (0.28–5.6 mg/kg) decreased SBP in diabetic rats. In line with our results, antihypertensive properties of NaSH at 0.56–5.6 mg/kg have been reported in animal models of hypertension [36,37,38,39,40] and in one study in type 2 diabetic rats [17]. H_2_S decreases blood pressure mainly by the opening of vascular K_ATP_ channels [41]; opening of these channels in pancreatic β-cells also decreases insulin secretion [4]. Our data indicate that effects of H_2_S on carbohydrate metabolism should be taken into account when favorable effects including its antihypertensive effect are intended.

Regarding strengths of this study, different doses of NaSH were administrated for a long time. In addition, an animal model of diabetes (HFD-STZ) which closely reflects characteristics of human type 2 diabetes was used; in this model, feeding a HFD induces insulin resistance and a low dose of STZ partially destroys β cells [42]. This study however has some limitations; serum total sulfide level was measured by the methylene blue method which measures all sulfur compounds rather than only free H_2_S; this method is however one of the most commonly used methods for measuring H_2_S in biological systems [43]. In addition, we did not measure oxidative stress and GLUT-4 protein levels in insulin sensitive tissues; these factors affect glucose metabolism in type 2 diabetes.

In conclusion, the unfavorable effects of chronic administration of NaSH in particular at high doses on carbohydrate metabolism in type 2 diabetic rats are reported. Further studies are indeed needed to clear the in vivo effects of different doses of H_2_S on carbohydrate metabolism in type 2 diabetes.

## 4. Materials and Methods

### 4.1. Animals and Induction of Diabetes

The care and use of laboratory animals was approved by the ethics committee of the National Institute for Medical Research Development (IR.NIMAD.REC.1396.307), Tehran, Iran. All animal treatments were carried out according to internationally approved guidelines for the use and care of laboratory animals. A total of 120 male Wistar rats (190–210 g) were housed under controlled conditions (23 ± 2 °C, relative humidity of 50 ± 6%, 12/12-h light-dark cycle). Rats had free access to water and a normal rat pellet diet with a total caloric value of ~3,100 kcal/kg, or a HFD with a total caloric value of ~4,900 kcal/kg.

Diabetes was induced using HFD-STZ as described previously [42]. After feeding the HFD for two weeks, a single IP injection of low dose of STZ (30 mg/kg dissolved in 0.1 mM citrate buffer, pH = 4.5; Sigma Aldrich, Hamburg, Germany) was done. One week after STZ injection, serum glucose levels were measured and rats with a fasting glucose levels ≥150 mg/dL were considered to be diabetic. The rats were allowed to continue on their respective diets for 9 weeks, when the study ended.

### 4.2. Experimental Design

Rats were randomly divided into 2 groups of 60 each: a control group and a diabetic group. Each group was sub-divided into 6 groups (n = 10) and treated daily with IP injection of NaSH (0.28, 0.56, 1.6, 2.8, and 5.6 mg/kg) or vehicle (normal saline) for 9 weeks. For minimizing oxidation of SH^–^ (hydrosulfide anion) the solutions were prepared and injected immediately (within 15 min). Body weight (using Tefal Scale; sensitivity 1 g), food intake (g/day), and water consumption (mL/day) were recorded every week. Fasting serum glucose concentrations were measured weekly for one month and then at the end of the study. Fasting serum levels of insulin and total sulfide levels were measured before and after the treatment. Intraperitoneal GTT was performed at the end of the eighth week of treatment and in vivo insulin secretion was assessed during GTT. PTT was done 3 days after GTT and blood pressure was measured at the end of the study. The experimental design is shown in Figure 9.

### 4.3. Measurement of Serum Glucose and Insulin Levels

After 12–14 h fasting, blood samples were collected from tip of the tail and centrifuged at 5000 g for 10 min. Serum levels of glucose were determined using commercially available kit (Pars Azmoon, Tehran, Iran). Intra- and inter-assay coefficient of variations (CVs) for glucose were 2.1 and 3.4%, respectively. A rat ELISA kit (Rat insulin ELISA; Mercodia, Uppsala, Sweden) was used for measuring serum insulin levels; Intra- and inter-assay CVs were 3.3% and 7.6%, respectively. The assay sensitivity was 1 µU/mL (6 pmol/L).

### 4.4. Measurement of Serum Total Sulfide

Total sulfide levels in serum were measured using the methylene blue method [44]. Briefly, 100 µL serum was added into a test tube containing zinc acetate (1% *w*/*v*, 200 µL), *N*,*N*-dimethyl-p-phenylenediamine sulfate in 7.2 M HCl (20 mM, 100 µL), and FeCl_3_ in 1.2 M HCl (30 mM, 133 µL). After incubation at 37 °C for 30 min, the tubes were centrifuged at 5000 g for 10 min. Supernatants were then collected for measurement of total sulfide. Total sulfide concentration was determined in the samples using a standard calibration curve established by 0–200 µM of NaSH at a wavelength of 670 nm using a microplate reader (BioTek, MQX2000R2, Winooski, VT, USA).

### 4.5. Intraperitoneal Glucose Tolerance Test, Pyruvate Tolerance Test, and In Vivo Insulin Secretion

For GTT and PTT, after 12–14 h of fasting, animals were anesthetized with an IP injection of sodium pentobarbital (60 mg/kg; Sigma Aldrich, Hamburg, Germany) and glucose (50% solution, 2 g/kg) or pyruvate (2 g/kg) was injected intraperitoneally. Blood samples for glucose measurements were collected from the tail vein immediately before, and at 10, 20, 30, 60, and 120 min after glucose or pyruvate administration [45]. Blood samples during GTT were used for assessment of in vivo insulin secretion.

### 4.6. Measurement of Systolic Blood Pressure

Systolic blood pressure was measured in anesthetized rats using a noninvasive tail-cuff method (AD Instruments, MLT125R, New South Wales, Australia) at room temperature as previously reported [46]. SBP values were averaged from three consecutive recordings obtained from each rat.

### 4.7. Statistical Analysis

GraphPad Prism software (Version 6, La Jolla, San Diego CA, USA) was used for data analysis. All values are presented as mean ± SEM. Two-way mixed (between-within) analysis of variance (ANOVA), followed by Fisher post-hoc test was used for analyzing data of water consumption, food intake, calorie intake, body weight, GTT, PTT, in vivo insulin secretion, and serum glucose. One-way ANOVA was used for comparing the area under the curves (AUC), SBP, and serum levels of insulin and H_2_S. Two-sided *p* values < 0.05 were considered to be statistically significant.

## Figures and Tables

**Figure 1 molecules-24-00190-f001:**
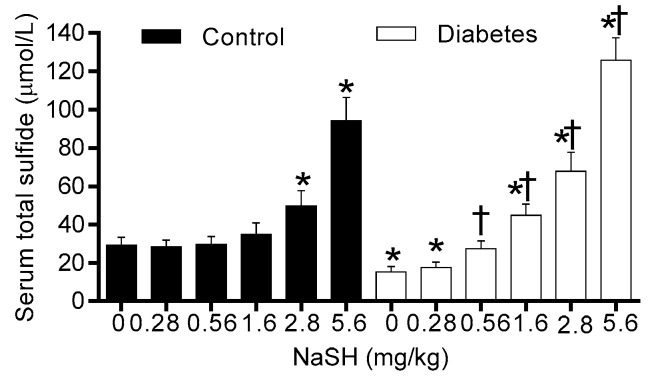
Effect of NaSH on serum total sulfide levels in control and diabetic rats. Rats were treated daily with different concentrations of NaSH for 9 weeks. Results are mean ± SEM (n = 10/group). * *p* < 0.05 compared to non-treated control rats, ^†^
*p* < 0.05 compared to non-treated diabetic rats.

**Figure 2 molecules-24-00190-f002:**
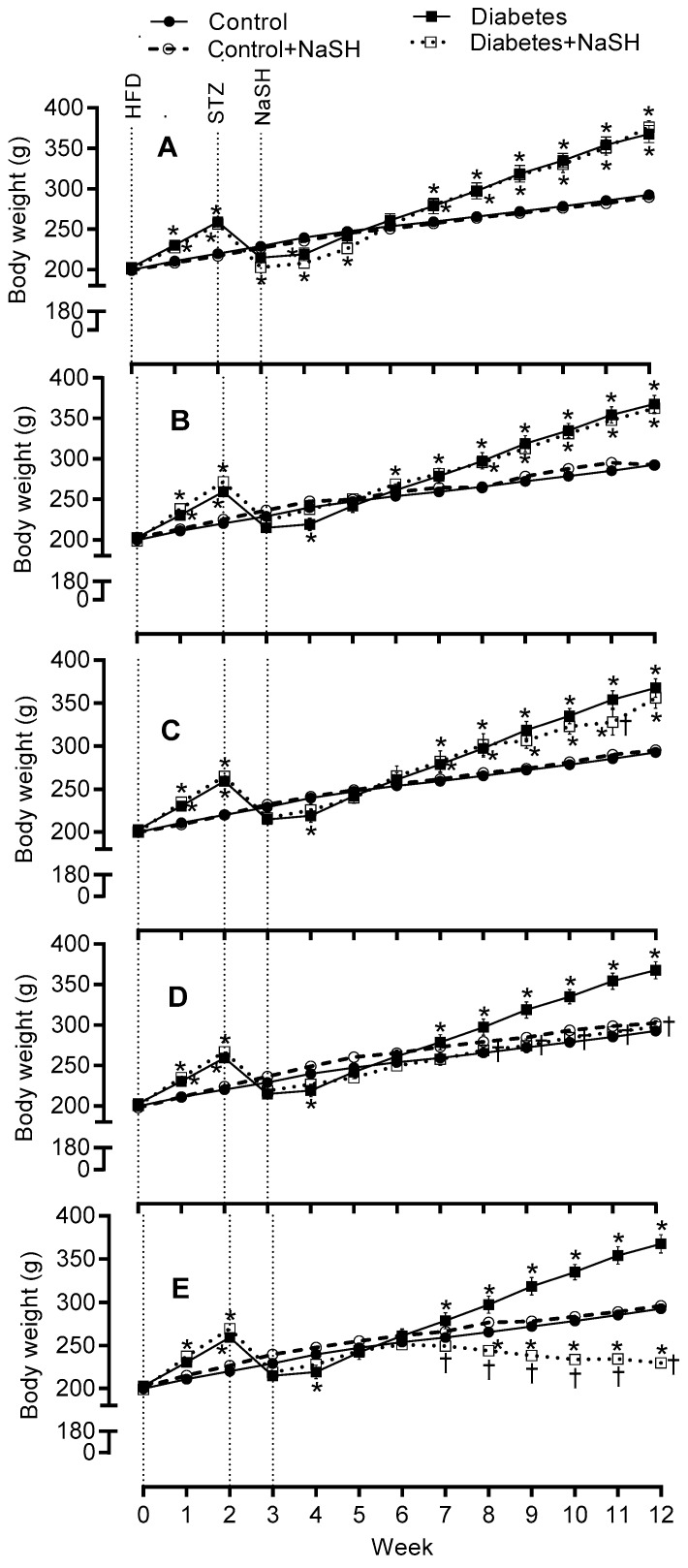
Effects of NaSH on body weight at 0.28 (**A**), 0.56 (**B**), 1.6 (**C**), 2.8 (**D**), and 5.6 (**E**) mg/kg. Results are mean ± SEM (n = 10/group). * *p* < 0.05 compared to non-treated control rats, ^†^
*p* < 0.05 compared to non-treated diabetic rats.

**Figure 3 molecules-24-00190-f003:**
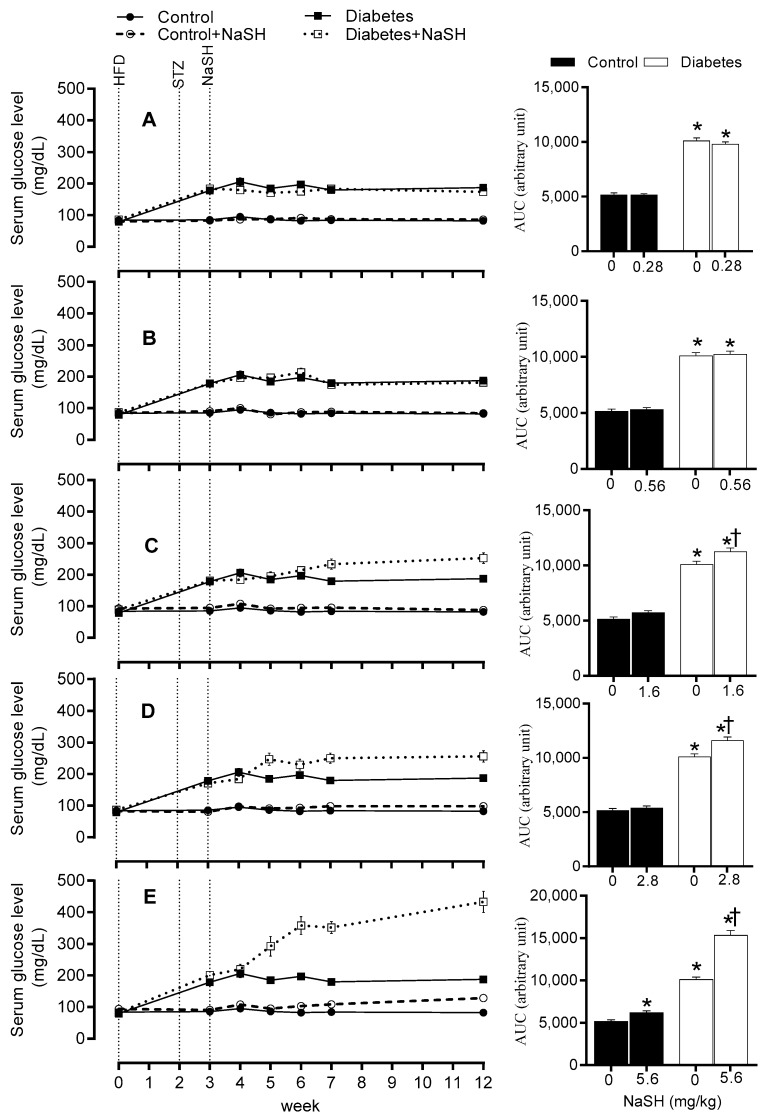
Effects of NaSH at 0.28 (**A**), 0.56 (**B**), 1.6 (**C**), 2.8 (**D**), and 5.6 (**E**) mg/kg on serum glucose concentrations. Area under the curves (from week 3 to week 12) are shown in columns on the right. Results are mean ± SEM (n = 10/group). * *p* < 0.05 compared to non-treated control rats, ^†^
*p* < 0.05 compared to non-treated diabetic rats.

**Figure 4 molecules-24-00190-f004:**
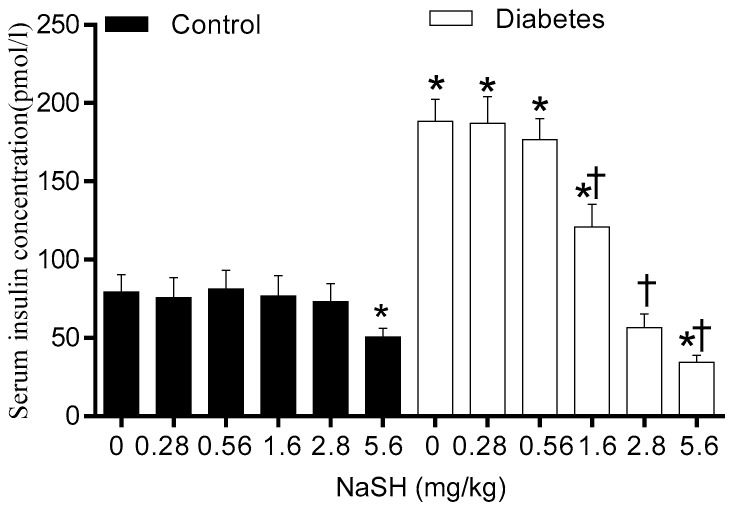
Effect of NaSH administration on fasting serum insulin levels at the end of the study. Results are mean ± SEM (n = 8/group). * *p* < 0.05 compared to non-treated control rats, ^†^
*p* < 0.05 compared to non-treated diabetic rats.

**Figure 5 molecules-24-00190-f005:**
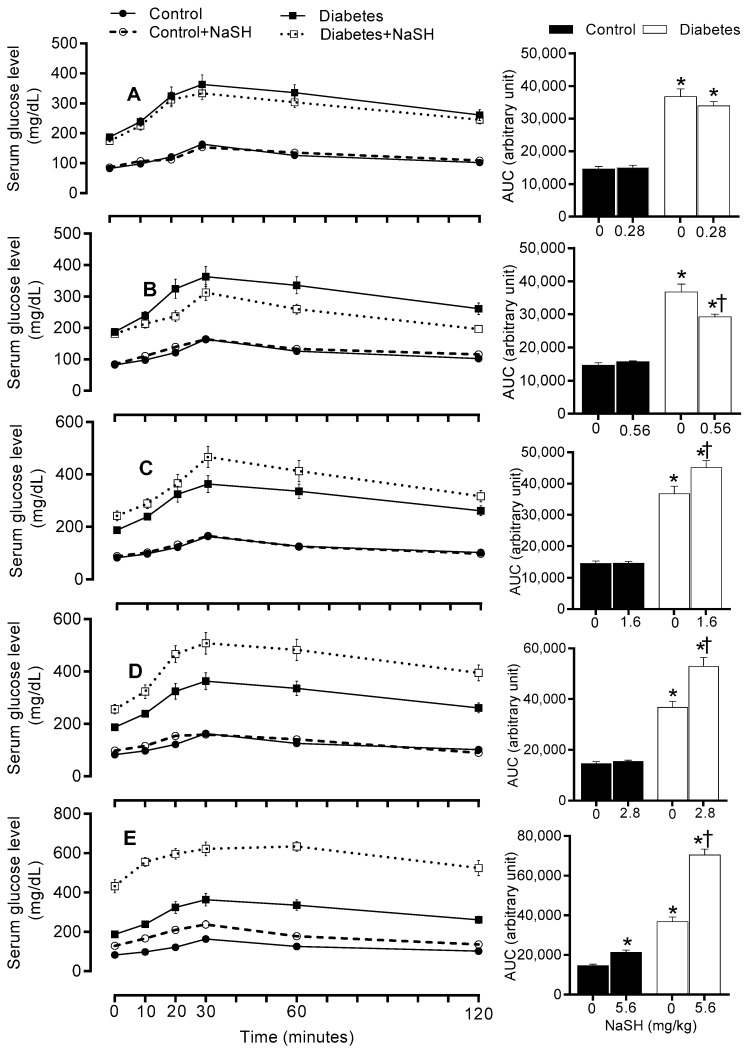
Effects of NaSH on glucose tolerance at 0.28 (**A**), 0.56 (**B**), 1.6 (**C**), 2.8 (**D**), and 5.6 (**E**) mg/kg. Area under the curves are shown in columns on the right. Results are mean ± SEM (n = 10/group). * *p* < 0.05 compared to non-treated control rats, ^†^
*p* < 0.05 compared to non-treated diabetic rats.

**Figure 6 molecules-24-00190-f006:**
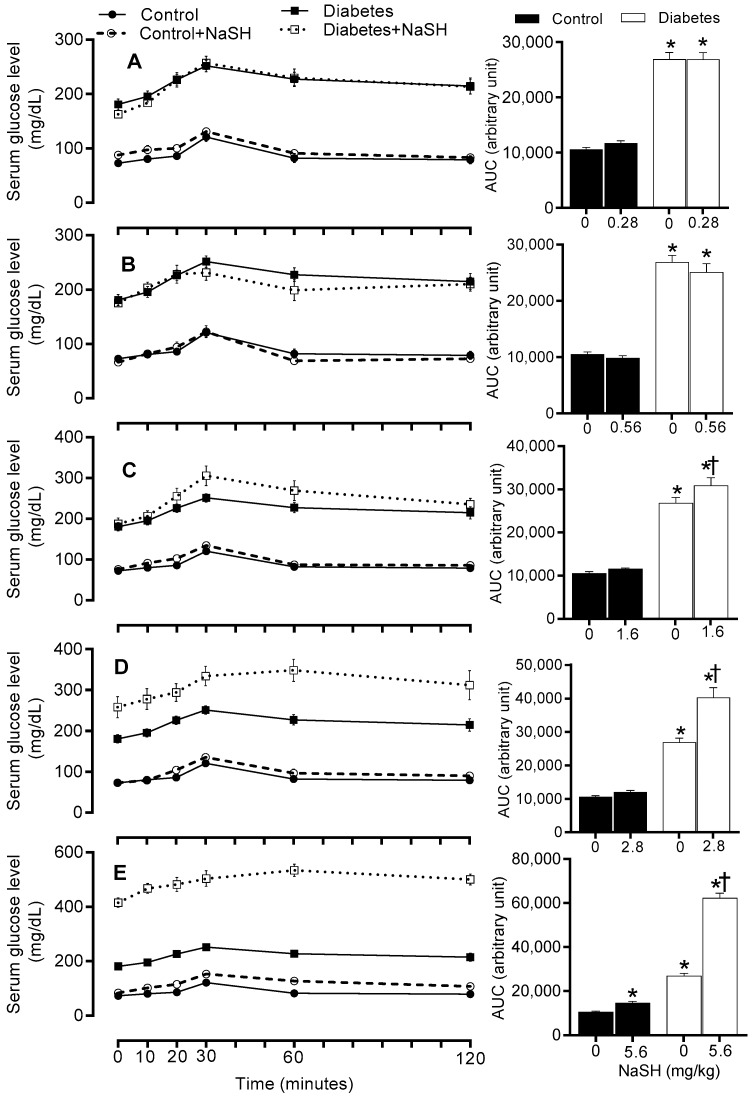
Effects of NaSH on pyruvate tolerance at 0.28 (**A**), 0.56 (**B**), 1.6 (**C**), 2.8 (**D**), and 5.6 (**E**) mg/kg. Area under the curves are shown in columns on the right. Results are mean ± SEM (n = 10/group). * *p* < 0.05 compared to non-treated control rats, ^†^
*p* < 0.05 compared to non-treated diabetic rats.

**Figure 7 molecules-24-00190-f007:**
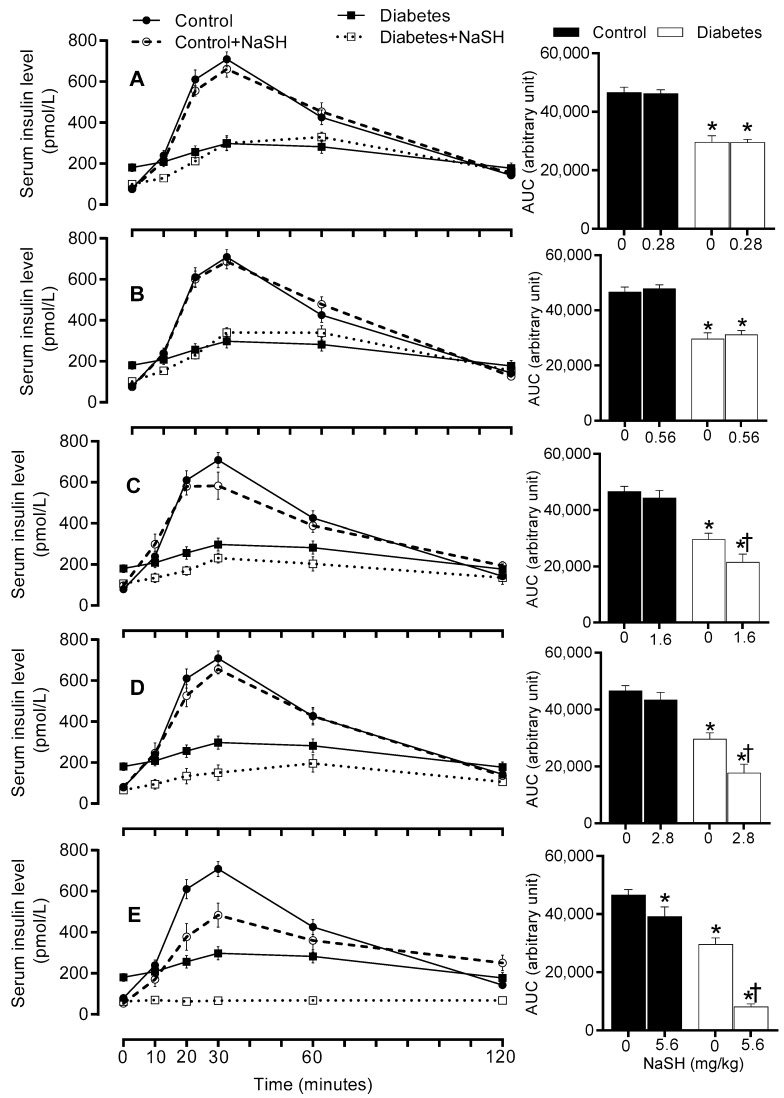
Effects of NaSH on in vivo insulin secretion at 0.28 (**A**), 0.56 (**B**), 1.6 (**C**), 2.8 (**D**), and 5.6 (**E**) mg/kg. Areas under the curves are shown in columns on the right. Results are mean ± SEM (n = 8/group). * *p* < 0.05 compared to non-treated control rats, ^†^
*p* < 0.05 compared to non-treated diabetic rats.

**Figure 8 molecules-24-00190-f008:**
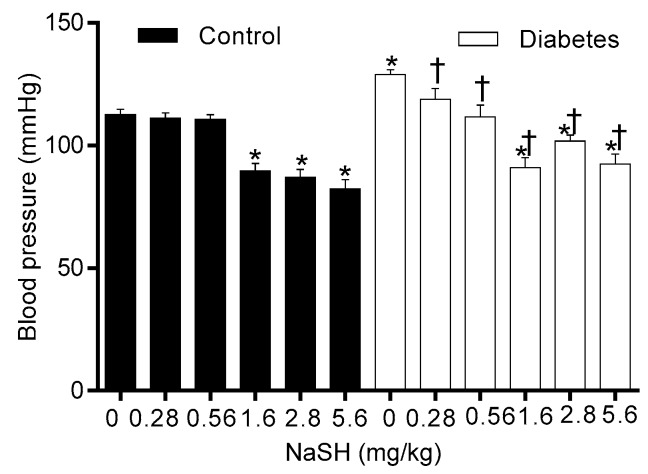
Effects of NaSH on blood pressure. Results are mean ± SEM (n = 9/group). * *p* < 0.05 compared to non-treated control rats, ^†^
*p* < 0.05 compared to non-treated diabetic rats.

**Figure 9 molecules-24-00190-f009:**
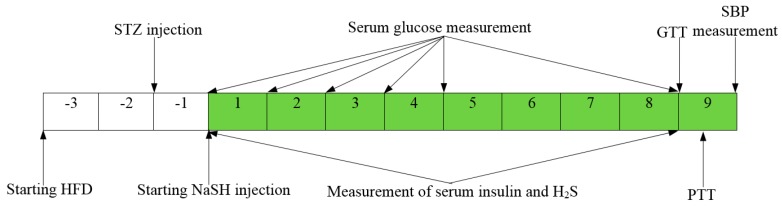
Experimental design; the dark squares indicate weeks of NaSH injection. HFD, high-fat diet; STZ, streptozotocin; GTT, glucose tolerance test; PTT, pyruvate tolerance test; NaSH, sodium hydrosulfide; SBP, systolic blood pressure.

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
