# Peer review of "Effects of Hydrogen Sulfide on Carbohydrate Metabolism in Obese Type 2 Diabetic Rats"

_molecules, 2019, doi:10.3390/molecules24010190_

Round 1

Reviewer 1 Report

The manuscript entitled “Effect of hydrogen sulfide on carbohydrate metabolism in obese type 2 diabetic rats” examined that the effects of hydrogen sulfide against diabetic rats that were treated with both high fat diet and low dose streptozotocin injection. Previous studies suggested that hydrogen sulfide showed positive or negative effects against diabetic animals. In this study, high dose administration of NaSH, a hydrogen sulfide donor, impaired carbohydrate metabolism in the diabetic rats.

The study provides important information of effects of hydrogen sulfide for treatment of diabetes. The results suggest that administration of NaSH has a possible to cause negative effects to diabetic patients. The results of between this study and previous reports were well discussed in discussion section.

Comment:

1). In abstract, the authors should describe that NaSH is a hydrogen sulfide donor.

2). 

The authors should check again in whole manuscript to improve the minor miss-descriptions. 

For example, 

Page 1, lane 19. In diabetic rats.NaSH at→ In diabetic rats, NaSH

Page 1, lane 21. Page 8, lane 142. Page 9, lane 185. Page 10, lane 237. in-vivo→ in vivo

Page 2, lane 54. in vivo→ in vivo

Page 9, lane 182. Page 10, lane 219. in vitro→ in vitro

Page 1, lane 23. Page 10, lane 235. H2S→ NaSH

All description about “mean ± SEM” in this manuscript should be fixed. For example, mean ± SEM.

Author Response

Reviewer 1

·         Comment: In abstract, the authors should describe that NaSH is a hydrogen sulfide donor.

·         Response: Agreed and done (page 1, line 14).

·         Comment: The authors should check again in whole manuscript to improve the minor miss-descriptions.

For example,

Page 1, lane 19. In diabetic rats.NaSH at→ In diabetic rats, NaSH

Page 1, lane 21. Page 8, lane 142. Page 9, lane 185. Page 10, lane 237. in-vivo→ in vivo

Page 2, lane 54. in vivo→ in vivo

Page 9, lane 182. Page 10, lane 219. in vitro→ in vitro

Page 1, lane 23. Page 10, lane 235. H2S→ NaSH

All description about “mean ± SEM” in this manuscript should be fixed. For example, mean ± SEM

·         Response: Agreed and corrected throughout the revised manuscript.

Reviewer 2 Report

Gheibi S et al investigated the effect of sodium hydrosulfide (NaSH) and H2S on glucose metabolism in healthy and high fat diet – streptozotocin (HFD-STZ) rats. The authors have found that NaSH application at higher doses disturbs glucose tolerance and insulin secretion, which lead to diabetes. The study is interesting, and the manuscript is well written. I got only minor comments:

In Fig. 2 at the first three weeks, the      body weights of diabetic animals increased and decreased due to HFD and      STZ injection compared to healthy controls. The authors calculated area      under the curve (AUC) for the complete 12 week period to determine the      effect of NaSH, which at high doses significantly reduced body weight      after long-term application. Since at the beginning of the experiments, the      body weight was altered due to HFD and STZ, and in the later stages due to      NaSH and long-term diabetic state, AUC calculation is not useful, since it      is not possible to distinguish among the “early effect” of HFD/STZ, and “late      effect” of diabetic state or NaSH effect. Therefore, it would be better to      omit AUC curves for Fig. 2 and to calculate significance for the single      values like comparing the effect of diabetic states and NaSH for each time      point one by one.

In Suppl. Fig. S3A, diabetic mice (denoted      by red circles) showed elevated water consumption 1 and 2 weeks after HFD      diet, however all other animals have similar values. Is this difference a      mistake or did these animals really drink more? If yes, why?

The time legend of x-axis for some figures      are not proportional (10 minutes=30 minutes=60 minutes) like for Fig. 5,      6, 7, Suppl. Fig. 12, 13.

Figure legends above the figures are not      uniform. It would better to use the same order of figure legends like as      it is used for Fig. 5 at the top left side: Control, Control+NaSH, right      side: Diabetes, Diabetes+NaSH.

The manuscript text, figure legends, legends      beside the figures, supplementary figure legends have some typos like “dianetes”,      “carve”, “arbitary unit”, “stroptozotocin”.

Author Response

Reviewer 2

·            Comment: In Fig. 2 at the first three weeks, the body weights of diabetic animals increased and decreased due to HFD and STZ injection compared to healthy controls. The authors calculated area under the curve (AUC) for the complete 12 week period to determine the effect of NaSH, which at high doses significantly reduced body weight after long-term application. Since at the beginning of the experiments, the body weight was altered due to HFD and STZ, and in the later stages due to NaSH and long-term diabetic state, AUC calculation is not useful, since it is not possible to distinguish among the “early effect” of HFD/STZ, and “late effect” of diabetic state or NaSH effect. Therefore, it would be better to omit AUC curves for Fig. 2 and to calculate significance for the single values like comparing the effect of diabetic states and NaSH for each time point one by one.

·            Response: Agreed; AUC curves for Figure 2 were removed and significance levels for the single values were added. In addition, in Figure S2 the AUC was calculated from week 3 to week 12 and this was stated in other figure legends.

·               Comment: In Suppl. Fig. S3A, diabetic mice (denoted by red circles) showed elevated water consumption 1 and 2 weeks after HFD diet, however all other animals have similar values. Is this difference a mistake or did these animals really drink more? If yes, why?

·               Response: Agreed; that was a mistake and corrected; thank you for your consideration.

·            Comment: The time legend of x-axis for some figures are not proportional (10 minutes=30 minutes=60 minutes) like for Fig. 5, 6, 7, Suppl. Fig. 12, 13.

·            Response: Agreed and done (Figures 5, 6, and 7 as well as Figures S11, 12, and 13).

·            Comment: Figure legends above the figures are not uniform. It would better to use the same order of figure legends like as it is used for Fig. 5 at the top left side: Control, Control+NaSH, right side: Diabetes, Diabetes+NaSH.

·            Response: Agreed and done.

·            Comment: The manuscript text, figure legends, legends beside the figures, supplementary figure legends have some typos like “dianetes”, “carve”, “arbitary unit”, “stroptozotocin”.

·            Response: Agreed and corrected.